# Equine-Assisted Intervention to Improve Perceived Value of Everyday Occupations and Quality of Life in People with Lifelong Neurological Disorders: A Prospective Controlled Study

**DOI:** 10.3390/ijerph17072431

**Published:** 2020-04-03

**Authors:** Anna María Pálsdóttir, Marie Gudmundsson, Patrik Grahn

**Affiliations:** 1The Department of Work Science, Business Economics and Environmental Psychology, Swedish University of Agricultural Sciences, P.O. Box 88, SE-230 53 Alnarp, Sweden; patrik.grahn@slu.se; 2Nature and Health, Region Dalarna, P.O. Box 712, SE-791 29 Falun, Sweden; marie.gudmundsson@regiondalarna.se

**Keywords:** equine-assisted therapy, animal assisted interventions, stroke, cerebral palsy, MS, disability, quality of life, occupational value, neurorehabilitation, horseback riding

## Abstract

People with neurological disorders suffer from poor mobility, poor balance, fatigue, isolation and monotonous everyday activities. Studies show that equine-assisted interventions can improve their mobility and balance, but could these kinds of interventions also increase participants’ activity repertoire and self-assessed health, and reduce their fatigue? The study was conducted as a prospective, controlled study with three cohorts followed for one year: intervention group (*n* = 14), control group Passive (*n* = 29), and control group Active (*n* = 147). Participants in the study were affected by neurological disease or injury that limited their opportunities for an active everyday life. The intervention group lacked regular activities outside the home before the intervention, which consisted of riding once a week, led by a certified therapist. Control group Passive lacked regular activities outside the home, while control group Active had several activities outside the home per week. Primary outcome measures were activity repertoire measured with Occupational Value Assessment questionnaire. Secondary outcome measures were global self-assessed health measured with EuroQol-VAS and fatigue measured with Shirom-Melamed Burnout Questionnaire. The intervention group’s activity repertoire and self-assessed health increased significantly compared to both baseline and the control groups. Equine-assisted interventions could help to improve the perceived value of everyday occupations and quality of life, as well as break isolation and increase the activity repertoire of people with neurological disorders.

## 1. Introduction

Since ancient times, relations with companion animals have been considered good for human health [1]. Not least, relations with horses. It is said that ancient Greeks offered people with incurable diseases horseback rides, as it was considered to improve mood [2]. The father of medicine, Hippocrates, talked about “riding’s healing rhythm” [3]. Ever since, horseback riding has been considered a method for improving both the mental and physical health of people. In fact, literature from the 17th century suggested riding as a treatment for e.g., gout and neurological diseases [4,5]. During the 18th century, contact with horses and other animals was introduced in psychiatric care [6]. Florence Nightingale was the first to officially record the value of animal assisted interventions since she deliberately used animals in therapeutic treatment in the mid and late 1800s [7]. Animals came to be used therapeutically in the care of soldiers with war trauma during the First and Second World Wars, and horses were used systematically in treatment and care in the 1950s [8]. A pioneer in the early 1950s was the Norwegian physiotherapist Elspeth Bödker, who treated children affected by polio, cerebral palsy and other neurological disorders with equine-assisted therapy. She found that the children did not get tired as quickly after undergoing therapy, and that riding gave them more change in the movements they needed to exercise, which she claimed led to faster improvements [9].

Since the 1960s, equine-assisted therapeutic activities have increased in scope and have become more and more professionalized. Activities have primarily grown in Germany and the USA [3]. Since the 1990s, more and more scientific studies have been conducted to investigate whether equine-assisted therapy works for different types of diseases and ill-health. Several systematic reviews have been conducted over the past ten years in animal-assisted and equine-assisted interventions. It is stated that animal-assisted interventions can improve mood and reduce depression in people affected by a variety of diseases, such as neurological disorders [10,11]. Equine-assisted therapy can improve both the physical and mental health of people with neurological disorders, such as injuries following road accidents and other types of physical trauma, as well as stroke, cerebral palsy, Multiple Sclerosis (MS) and many other types of neurological disorders. It is a matter of improving balance [12,13,14], movement [15,16] and walking [14,17]. In addition to physical injuries, the interventions also seem to improve mood and recovery from exhaustion [18]. A study by Cabiddu et al. [19] shows that equine-assisted therapy had a calming and restorative effect as measured by heart rate variability (HRV) autonomic regulation.

The studies thus show promising results in terms of improvements in both physical and mental health. However, there are some criticisms of the studies being done in the field, i.e., that they do not provide sufficient support to be able to draw any reliable conclusions about the physical or psychological improvements of equine-assisted therapy for people with neurological disorders [20]. That is because most studies lack controls. Moreover, the studies that have controls are small, and the controls often consist of people who do not have neurological disorders. Many studies only have short interventions and are lacking in follow-ups. Also, descriptions of the interventions are often lacking or incomplete. Among other things, many times there is no indication of whether the intervention is led by professional therapists or not [21,22,23,24]. In addition, animal ethics are all too often overlooked [25,26].

Although scientific evidence does not yet clearly show whether and how equine-assisted interventions affect the state of health and functions for people suffering from neurological disorders, many studies show that this type of therapy can improve participants’ balance, mobility and mood. However, health and well-being are not just about symptoms associated with the neurological diseases per se. In general, it is about not being isolated in time and space, about being able to have social contact and about getting structure in everyday life. It is about being able to have an everyday life that contains activities that bring meaning and context in life [27,28]. Whether equine-assisted activities can influence this can be investigated both through interviews and by measuring participants’ everyday activity repertoire. How does the everyday life of people with neurological disorders change if they can participate in equine-assisted interventions? Can they broaden their repertoire of activities? Do they feel that health has improved? Do they feel more alert and energetic? To our knowledge, such a study has never been conducted before.

### Aim

The aim of the study was to investigate the effects of an equine-assisted intervention on the participants’ perceived value of everyday occupations and health as well as to gain deeper understanding of what the interventions can mean for participants in everyday life. 

## 2. Materials and Methods

The study was conducted as a prospective, controlled, before-and-after study [29,30,31]. In intervention studies in e.g., public health and rehabilitation, where the alternatives cannot be blinded to the participants, randomized controlled trials might not be the best choice [32]. Challenges include ethical problems and dropouts, especially if the intervention is considered more beneficial than the alternative [33,34,35]. In a case such as this, when a randomized controlled trial (RCT) is not feasible, we found advantages in including two nonrandomized control groups instead of relying solely on a pre- and postintervention comparison. Our control groups helped us to account for threats to internal validity (e.g., regression to the mean). The use of a controlled before-and-after study also reduced threats to external validity, which often limit the value of RCTs. RCTs are normally conducted in certain highly selected and special settings and are seldom done at ordinary community sites. By using this type of design, we could involve common riding schools, thereby making the results more generalizable. The lack of randomization often facilitates recruitment of a larger proportion of eligible participants, further increasing generalizability [36]. An option that we chose is natural experiments, where people are followed in their daily lives. We chose to follow three cohorts, all of which have similar participants in terms of age, gender and diagnosis, but where the difference was in how often they have activities outside the home. The control groups live their lives as usual, where there is no question of any kind of intervention. In this way, the relationship between the intervention group and the controls can be described as a natural experiment or a quasi-experiment [37,38].

We submitted the study for ethical review at Lund University. They advised us to gather and provide all participants with informed consent in accordance with the WMA Declaration of Helsinki, which we did [39]. Among other things, the participants were informed that they could end their participation in the study at any time without having to give reasons. Each participant was asked for their written consent to participate in the study, which was gained from all included participants.

We chose to describe the intervention as well as to measure the effects and give a description of how the intervention was experienced by the participants. The study therefore applied a mixed model design to give the possibility of a broad and deep analysis [40], including both quantitative and qualitative analyses. The evaluation was conducted as a prospective longitudinal controlled study that included validated instruments for an intervention group and two control groups. Furthermore, a retrospective interview study was planned with former participants in the intervention.

### 2.1. Setting

The project started in 2012 and took place in the county of Dalarna, which is located in the central part of Sweden, northwest of Stockholm. The intervention included four riding schools in four towns in Dalarna. The goal of the intervention was to help people with neurological disorders to increase their access to equine-assisted interventions within the existing system of public health work and healthcare, and this goal is met today. The project was run by the Neuro Association, with the support of the Swedish Equestrian Federation, the Dalarna County Council and the municipalities of Dalarna.

### 2.2. Intervention

Approach: The target group in the project was adults with neurological disease or injury. These adults lack regular activities that include recreational activities involving physical activity. It is a target group that, to a great extent, experiences poorer health than the population on average and that needs lifelong recurring rehabilitation. The project offered equine-assisted interventions; a targeted exercise individually in groups. The intervention we carried out can be defined as “Animal-Assisted Activity” according to the International Association of Human–Animal Interaction Organizations (IAHAO) [41] and thus can be defined as Equine-Assisted Activity (EAA) according to affiliated organizations, such as the Professional Association of Therapeutic Horsemanship International (PATH International) [42]. Working with the horse and the surrounding physical and social environment, the therapist creates a supportive environment [43,44,45] that aims to promote and develop the participant’s health, social function and/or learning, thereby contributing to an active and meaningful everyday life from a holistic perspective. Equine-assisted activities are intended to affect body functions, such as balance, coordination and fitness; cognitive functions, such as memory and space perception; psychological functions, such as self-confidence and mood, and social functions, such as interacting with others in a group. This is done through interaction with the horse on its back or in activities in the stable and in the care of the horse [1].

Team: The activity was led by a licensed occupational therapist who was also certified in equine-assisted therapy (hereinafter called the therapist). The aim of the therapist’s work was to support the participant’s capacity for activity and participation in society in a way that facilitates the opportunities to live as valuable and good a life as possible. The efforts were based on the person’s own view of their situation and their needs and considerations of opportunities and obstacles in the environment. This therapist also guided a physiotherapist who led some of the groups, as well as a riding instructor and assistants from each participating riding school in the project. Throughout the intervention, the leaders followed the participants’ individual goals for the activity. There was also collaboration with a rehabilitation clinic in the county.

Length of intervention and periodicity: The intervention included one weekly session for a total of twelve months. In total, the participant stayed between one and one and a half hours per session in the stable, but the active riding session was a maximum of 30 min.

The intervention: The first riding event for each participant was done individually to provide the opportunity for the therapist and the participant to create a calm and trusting contact, to assess the status and the opportunity for equipment adaptation, and for an initial match between rider and horse. Both riders and horses need to thrive and work well together to achieve success. The participant then entered into regular activities in a smaller group.

At each session in the regular activities, the participants arrived at their respective start times. Most participants came a while before and had time to talk to the other participants who had ridden or were going to ride. Some helped to prepare the horses both before and after riding. It was not a requirement, but could be included for those who wanted. The ride started by mounting the horse, using stairs, ramp or ceiling lift. An assistant and/or the riding instructor assisted.

Participants rode in small groups of 2–4 people. Each riding session on the horse lasted for a maximum of 30 min but could be as short as about 10 min and was adapted to each participant’s ability and current state. Assistants led the horses to the extent agreed between the therapist and participants. Initially, the participants were instructed to focus on finding their balance, relaxing and following the horse’s movements. This could involve breathing exercises, short relaxation exercises and simple body awareness exercises. This was followed by different exercises where the participant more actively interacted with the horse, such as riding over bars, riding slalom between cones, changing tempo, stopping and turning. During trail rides outdoors, these exercises occurred naturally in the forest, where the substrate and surrounding environment vary and provide different types of challenges and experiences. Before completing the exercise, the participant finished by relaxing and focusing on feelings in their own body. The exercises were reasonably consistent, but were adapted to the different participants’ daily state and fitness. The riding usually took place indoors, but was moved outdoors when weather permitted. What was defined as “good weather” depended on the participants’ own perception. Some felt well when it was warm, others when it was cooler.

### 2.3. Study Population

#### 2.3.1. Intervention Group

Recruitment of participants was done partly through the Neuro Association at the local level, and partly through healthcare professionals and municipal staff who were informed about the project.

Inclusion criteria: Adult inactive individuals with neurological disease or injury who lived in the county were invited to participate in the intervention. Participation was voluntary.

Exclusion criteria were assessed based on both the horse’s well-being (e.g., the participant’s weight being over 100 kg) and the participant’s health and function (e.g., that the participant had very poor body balance so that they could not sit on the horse without a helper on either or both sides; or had epilepsy that was not stably medicated).

A total of fifteen people were selected to be included, of whom 14 completed the study: 12 women and 2 men. The age ranged from 22 to 71 years at inclusion. Participants had neurological disease or injury with varying symptoms such as cognitive difficulties, pronounced mental fatigue, pain and more or less impaired mobility. Diagnoses represented in the participants include MS, stroke, muscular disease, polyneuropathy, fibromyalgia and cerebral palsy.

Ten of the participants had progressive diseases. Many were dependent on mobility aids and assistance. Nine participants had walking difficulties and walked with some difficulties, or walked slower than normal, or (3 people) were fully wheelchair bound. Eight participants had balance difficulties and therefore had trouble being independent in movements and standing, which causes difficulties in independently performing everyday activities standing. This can affect actions such as dressing, hygiene and toilet visits, cooking/baking, cleaning, etc. Seven participants had problems with extreme fatigue, which means having reduced endurance in activities, becoming exhausted and needing rest. This relates to a kind of fatigue that is not due to lack of sleep and that leads to many problems in everyday life. It can be in the form of difficulty in completing activities, uncertainty about being able to participate in social contexts and reduced ability to focus attention.

#### 2.3.2. Control Groups

Members of the Neuro Association, all suffering from various neurological disorders, were invited by email to participate in the evaluation. Participation was voluntary and anonymous. We defined control groups according to Carter and Lubinsky [46]: “active control group” means that participants receive standard treatment, treatment as usual or participate in an activity that according to common practice gives effect. “Passive control group” gets placebo or no treatment at all. In total, 245 participants agreed to participate in the “Active” control group (active one or more times a week; organized or self-organized activities such as walking or gym training on their own or with a coach) and 40 participants in the “Passive” control group (had no activities during the week). In total, 147 respondents from the Active group (98 missing) and 29 respondents from the Passive group (11 missing) responded on all three occasions and were thus included in the analysis.

At the start of the intervention as well as after 6 and 12 months of inclusion, the participants in the intervention group had to complete a number of self-assessment forms regarding health and function. The control group responded to corresponding surveys in the form of web surveys that were sent out via the Neuro Association. A reminder was sent out once a week after mailing.

Table 1 shows that the differences between the groups are small in terms of gender and age. Due to their participation in several activities per week outside the home, we assume that participants in the Active control group, have more resources and thus have a greater range of activities in their everyday lives. One assumption is that they also have better values for other outcome variables. We assume that participants in the Passive control group should have the same baseline values as the participants in the intervention group for all outcome variables. The question is whether one single activity per week for the participants in the intervention group can change the values of the outcome variables in relation to the participants in both the Active and Passive control groups.

### 2.4. Quantitative Measures

Prior to the start of the intervention, baseline data was collected by asking participants to complete a number of self-assessment forms regarding health and activity repertoire. Follow-up data was collected 6 and 12 months after the start. The control group responded to corresponding surveys in the form of web surveys sent out via the Neuro Association. A reminder was sent out a week after the web surveys were published.

The primary outcome measure was perceived value of everyday occupations. Occupational Value Assessment (OVal-pd) is a validated and reliability-tested instrument [47]. It consists of 18 statements concerning the perceived value of everyday occupations that the participant has performed within the past month. Four response alternatives are possible for each statement: not at all (1), rather seldom (2), rather often (3), and very often (4). The statements are related to three dimensions: concrete (visible product), symbolic (individual and indirect value) and self-rewarding value (the activity itself is rewarding). The instrument is based on the ValMO model (The Value and Meaning in Occupations). It emphasizes the importance of having a balance between work, play, leisure and self-care/maintenance activities. These activities have an individual value for the person and how they experience the activity. It is also important that the person experiences a balance between the three dimensions of concrete, symbolic and self-rewarding [48].

#### Secondary Outcome Measures Were the Following

Shirom-Melamed Burnout Questionnaire (SMBQ): This is a self-assessment instrument that measures exhaustion and is related to long-term stress [49,50]. SMBQ has been used in Swedish studies and has been shown to have high validity and reliability [51]. The Shirom-Melamed Burnout Questionnaire consists of four subscales: emotional and physical exhaustion (eight items); cognitive weariness (six items); tension (four items) and listlessness (four items). All items have a seven-point response scale (1 = almost never, 7 = almost always).The tested persons were asked to estimate how well their experience matches the statements in the instrument using a seven-point scale, where 1 = “almost never” and 7 = “almost always”. The average of each dimension is a measure of burnout, and a high average is equivalent to a worse condition. It is proposed that the clinical limit value should be 4.47, which means that those above the value are at risk of mental exhaustion [51].

The SMBQ dimension listlessness contains statements such as: “I feel full of energy”, “I feel alert”, “I feel active”, “I feel drowsy”. The dimension has to do with desire, will, zeal and motivation.

The SMBQ dimension tension is about physical tension and contains statements such as: “I’m tense”, “I feel a strong inner tension”, “I feel restless”.

The SMBQ dimension emotional and physical exhaustion contains statements such as: “I feel tired”, “I feel physically exhausted”, “I feel I’ve had enough”, “My ’batteries’ are depleted”, “I feel burnt out”, “I feel mentally tired”. The dimension contains essential statements related to mental exhaustion.

The SMBQ dimension cognitive weariness contains essential statements related to fatigue. The dimension contains statements such as: “I feel tired in my head”, “I have trouble concentrating”, “It feels difficult to think about complicated things.”

EuroQol-VAS (EQ-VAS): This is a global measure of self-assessed health. Respondents were asked to mark their perceived health status on the day [52] using a 20 cm vertical scale with end points of 0 (at the bottom) and 100 (at the top). The respondents were asked to put a cross on the scale, where 0 = “the worst health you can imagine” and 100 = “the best health you can imagine”. The normal value for a population is 84.0 according to EuroQol [52]. A Swedish study [53] shows that the normal value for a young population, aged 18–24, is 84.3. EQ-VAS is used extensively, e.g., when researchers compare the status of quality of life [54] or in Health Economics [55]. EQ-VAS is considered to be reliable and valid [54] as well as sensitive [56].

Statistical analyses: To analyze the hypothesis that the change over time is zero as well as that the change over time is equal between the groups (intervention/control), ANOVA SAS General Linear Model, Proc GLM was used [57,58], including Tukey’s test [59]. SAS software (release 9.4m6, Cary, NC, USA) was used for analysis, and a significance level of 0.10 was applied. In this case, we have good knowledge of all persons included in the study. We also have qualitative analyses that can back up the quantitative methods. The amount of information for each participant is high, which means that we can chose the significance level *p* < 0.10 [60,61,62,63]. Where relevant, the significance level *p* < 0.05 is also reported.

*p*-values provide limited information on the clinical efficacy [64,65]. We therefore chose to report the effect size together with the *p*-values. We calculated the effect size [66] based on differences between means (intervention group and control) using Hedges’ g [67]. Hedges’ g is proposed when the intervention group is 20 or less. An effect size >0.80 compared to control is explained as high; >0.50 is explained as medium and >0.20 as discernible.

### 2.5. Qualitative Methods

Five semi-structured interviews were conducted after asking six people about participation via email. One participant abstained. The selection was stratified based on variations of the participants related to gender, age, diagnosis, whether they worked or not, and whether they continued to ride after the project was completed or not.

A time was decided, and the participants chose the location of the interview. Two interviews were conducted in the home and three at another location.

The interviews were conducted individually and each one took about an hour to complete. Written consent was obtained before the interview, which was recorded and then transcribed. Participation in the study was anonymous and the results are presented at group level.

The transcribed texts were analyzed according to Interpretative Phenomenological Analysis [68] to shed light on the participants’ perspectives on their participation in the intervention.

## 3. Results

The quantitative results will be presented first, followed by the qualitative analysis.

### 3.1. Quantitative Measures

The results are presented as follows: primary outcome measure, secondary outcome measure, and then the results of the interview study.

#### 3.1.1. OVal-pd

OVal-pd is presented as three dimensions: (1) concrete (visible product), (2) symbolic (individual and indirect value) and (3) self-rewarding value (the activity itself is rewarding).

OVal-pd concrete value: The value of the intervention group (Figure 1) increased by 0.28 units when baseline was compared to a 12-month follow-up (*p* = 0.0630). Compared to the Passive control group, the value is very clear (*p* = 0.0310). Concrete value means that an activity is done to achieve a concrete result. The study indicates that the participants in the intervention perceived that a concrete result is achieved with this equine-assisted activity, such as developing previous skills or learning new ones.

The value decreases over time for the Passive control group, while Active group remains unchanged over time, i.e., neither new nor previous skills appear to have developed. The values are perfectly reasonable, i.e., you can expect that the Active group will be higher than the Passive.

Effect size: The intervention group increased significantly over time and passed the Active control group. The effect size is medium after 12 months for comparisons between the intervention group and Active control group, where Hedges’ g is 0.54. The effect size is nearly high for comparisons between the intervention group and the Passive control group, where Hedges’ g is 0.74.

OVal-pd symbolic value: The value of the intervention group increased by 0.32 units when baseline was compared to a 12-month follow-up (*p* = 0.0896). The values for the control groups, Active (Figure 2) and Passive, follow essentially the same pattern. There is a clear difference between the intervention and the Passive control group (*p* = 0.048 ANOVA). A follow-up check with Tukey analysis shows a slightly higher value (*p* slightly above 0.05). The results indicate that the participants feel that they are performing activities where they feel increased symbolic values. It may be that they feel they are part of a larger social and cultural context and their self-esteem has increased.

Effect size: With respect to symbolic value, the value of the intervention group also increased significantly with time and passed the Active control group. The effect size is medium after 12 months. Hedges’ g is 0.65 when comparing the intervention group and the Active control group, and also 0.65 when comparing the intervention group with the Passive control group.

OVal-pd self-rewarding value: The value of the intervention group (Figure 3) increased by 0.37 units when baseline was compared to a 12-month follow-up (*p* = 0.0688). The difference is greatest in the 12-month follow-up between the intervention and the Passive control group (0.37 vs. 0.03, *p* = 0.088). The results show that the participants in the intervention group perceive that they perform more activities that they experience as self-rewarding, i.e., that they do something that they perceive as really exciting and that their skills match the challenge of the activity.

Effect size: The value of the intervention group increased over time and passed the Active control group. Both control groups (Figure 3) are relatively the same over time. The effect size after 12 months, measured by Hedges’ g, is medium: 0.61 for comparisons between the intervention group and the Active control group, and 0.60 for comparisons between the intervention group and the Passive control group.

#### 3.1.2. Shirom-Melamed Burnout Questionnaire (SMBQ)

This instrument consists of 22 statements related to the following four different dimensions: (1) listlessness, (2) tension, (3) emotional and physical exhaustion and (4) mental fatigue. These are presented in figures comparing the intervention group (red) with the control groups (blue and green).

SMBQ listlessness: For the SMBQ dimension listlessness, values continuously improved for the intervention group (Figure 4) compared to the control groups, i.e., listlessness decreased steadily, but the differences are not significant. This dimension has to do with desire, will, zeal and motivation, which is in line with what is dealt with in OVal-pd, all dimensions, but not least when it comes to self-rewarding value.

Effect size: The intervention group showed steady improvement and is almost at the level of the Active control group. The effect size measured by Hedges’ g after 12 months is noticeable: 0.22 for comparisons between the intervention group and the Active control group, and likewise 0.22 for comparisons between the intervention group and the Passive control group.

SMBQ tension: For the SMBQ dimension tension, the intervention group’s (Figure 5) results were considerably improved between baseline and follow-up, especially between baseline and six-month follow-up. These statements deal with perceived stress, which affects both mental tension and tension in the musculature. Although the results are considerably improved compared to the control groups—especially compared to the Passive control group—the values are not statistically reliable at 12-month follow-up.

SMBQ emotional and physical exhaustion: The SMBQ dimension emotional and physical exhaustion contains essential statements related to exhaustion. The intervention group (Figure 6) improved capacity after 12 months. However, the values do not differ significantly from the control groups.

Effect size: What distinguishes the intervention group is that, above all, the values improved between six and twelve months, i.e., at the same time as the tension and stress began to drop. The effect size after 12 months, measured by Hedges’ g, is noticeable: 0.30 for comparisons between the intervention group and the Active control group and only 0.07 for comparisons between the intervention group and the Passive control group.

SMBQ Cognitive weariness: The SMBQ dimension cognitive weariness did not improve with the intervention (Figure 7); quite the opposite, which probably has to do with riding involving the introduction of a new activity that—if one is not familiar with—places high demands on concentration and can lead to cognitive weariness. However, the increase is not great, and the difference compared to the control groups is not significant.

Effect size: The three groups (the intervention group and the two control groups) are in parallel for the first six months. Thereafter, the Passive control group decreases, while the values primarily for the intervention group continue to rise slightly. The effect size at the 12-month follow-up, measured by Hedges’ g, is low, 0.14, for comparisons between the intervention group and the Active control group, and discernible, 0.27, for comparisons between the intervention group and the Passive control group.

#### 3.1.3. EQ-VAS

For the intervention group (Figure 8), the global self-estimated value rises forcefully and significantly (*p* < 0.05) between the baseline value (54.0) and the six-month follow-up (66.0). The intervention group is the lowest of the groups at the start. The change over time differs compared to the control groups and is significant (*p* = 0.03), in particular compared to the Passive control group (*p* = 0.02). However, all values are below the normal value 84.0.

Effect size: The effect size after twelve months, measured by Hedges’ g, is medium, 0.59, for comparisons between the intervention group and the Active control group, and nearly medium, 0.46, for comparisons between the intervention group and the Passive control group. At the follow-up after six months, the figures were higher, where the effect size was medium, 0.71, for comparisons between the intervention group and the Active control group, and medium, 0.68, for comparisons between the intervention group and the Passive control group.

### 3.2. Qualitative Analysis

The interviews were analyzed using Interpretative Phenomenological Analysis [68], which resulted in four main themes.

1:Strengthening and promoting abilities and well-being2:Increased empowerment3:Equality and justice through increased accessibility4:The horse and its surroundings as a resource

#### 3.2.1. Theme 1: Strengthening and Promoting Abilities and Well-Being

This theme describes difficulties in everyday life that are associated with the diseases, such as having poor balance, walking ability, energy and a poor ability to focus attention. Many suffer from fatigue due to insomnia and exhaustion, are depressed, and have poor self-confidence. Many participants have progressive diseases and are therefore experiencing a worsening condition.

The theme contains many depictions of how participants notice that health has improved by participating in the intervention. It deals with a wide range of factors, such as better balance, strength, energy, quality of sleep and, not least, improved self-confidence. They feel generally happier and more satisfied with both their physical and mental health. Especially positive is that participants with progressive diseases claim that they feel that the deterioration of their health is being slowed down. They generally feel more excited about the exercises, which both take and give strength.

#### 3.2.2. Theme 2: Increased Empowerment

The participants describe how difficult it is for them to manage ordinary, everyday activities independently, such as moving about and managing the household. In addition to all the activities in the homes that they have to undertake, they experience great difficulties in finding suitable and meaningful activities outside the home. It is difficult to be physically active since the selection of possible activities is limited due to, for example, poor availability and adaptation of premises. They say that they therefore do not experience challenges in a positive sense. In general, this means that they do not have routines outside the home, which means that they can easily become isolated and alone. Much could also be improved if their finances were generally not so bad.

This limits the possibilities of things such as travel and private purchases of, for example, a horse.

This theme also indicates that participation in the intervention has meant several perceived improvements in the participants’ health and well-being. As a result, they generally endure much more in everyday life. Above all, they report that the intervention has significantly strengthened their perception of what they can actually do. Their confidence in being able to cope and master situations themselves has increased significantly. As a result, their empowerment has increased substantially.

Through riding, they have found a meaningful activity that is both exercise and leisure. They feel motivated to participate for a really long time in riding. They state that this activity, unlike others they have participated in, matches and challenges their abilities and can be tailored to their particular needs. They thereby feel less pressure and fewer performance requirements. Instead, they can more fully enjoy the activity and at the same time experience a sense of freedom. They feel selected and seen, healthy, brave and capable.

The intervention means that they have a changed role in that situation. They feel freer from their illness, going from defining themselves as patients to becoming riders. In that situation, they may feel like “anyone”, a rider among other rider friends. They are people who are able to exercise and be active.

A very important aspect of the intervention has been that it has led to them being put in a social context. They meet friends outside of healthcare, where they have fun and can laugh together. They can share an interest with others, which creates a positive community.

#### 3.2.3. Theme 3: Equality and Justice through Increased Accessibility

Hardly anyone in the group has the financial conditions to be able to ride as they wish with the costs it entails. They wish they could ride more than once a week, but cannot afford to pay for private riding with the help that is needed.

Participants describe the mixed feelings they gained from being able to participate in the intervention. There was a desire to be selected as a participant in the project, while also creating a sense of injustice, as there are many others who would like to and have a need. It should not be a cost issue and should not cost more than for other leisure riders. Politicians should understand that there is a great benefit to patients being able to ride, in spite of their needs and barriers in everyday life, and that it should then be possible for patients to continue riding as self-training and health promotion. It is a significant benefit to society that citizens can maintain their health, despite a continuing illness.

#### 3.2.4. Theme 4: The Horse and Surrounding Environment as a Resource

Accessibility: Participants say that they have a great need to be in environments that are not stressful, but that give them peace and quiet. Places they can be in without discomfort should be free of disturbing sounds. The environments must be accessible at the same time. With regard to this intervention, there must be a lift and ramp.

Personnel: The intervention worked well, much because it was performed by staff who were experienced in providing a safe atmosphere. The staff is known in the area for being knowledgeable.

The horse: The intervention was performed with safe and calm horses. Initially, time was given for the participants to try different horses so that they were specially selected as a suitable fit for the rider. Participants described that the role of the horses changed during the intervention, from being a training tool to becoming a creature with which to interact. It was nice to spend time with the horse, soothing.

The interaction meant silent communication. The horse helped participants with body movements when it was difficult for them to move themselves. It also helped the participants with relaxation and was perceived to give a feeling of happiness.

Stables and riding hall: These give a good impression. Sometimes there is too much noise and talk. The focus is on training, which is good. There is a possibility, but not a requirement, to help.

Nature: All participants prefer to ride outside. Through riding, nature becomes accessible; you can get to places that would otherwise be impossible to get to. Riding in the outdoors opens up possibilities for excitement and surprises; it becomes a special challenge. It makes it possible to get close to both the horse and nature at the same time, without effort and without thinking about the fact that it is exercise. It is perceived as freer than riding indoors, where the riding session becomes exercise, as riding outside offers mental recovery and rest. Riding in the outdoors reinforces a here-and-now experience; it becomes a kind of mindfulness. It becomes easier to know the body, and at the same time to interact with the horse and the surrounding nature.

## 4. Discussion

The results indicate that equine-assisted activities have the potential to be health-enhancing activities that contribute to maintaining and/or improving the health of the individual in terms of function, activity and participation.

In the study, we had two types of control groups: Passive, where the participants did not perform any activity at all during the week and Active, where the participants had some type of activity up to four times each week. The Passive control group consists of 29 persons and the Active control group consists of 147 persons. With such a small number in the intervention group (*n* = 14), the changes must be very large and tangible, with little spread in the group, in order for the change to be classified as statistically significant.

The results show that the changes between baseline and follow-ups in all three dimensions of the OVal-pd are statistically significant (*p* < 0.10). The primary outcome variable, OVal-pd, thus increases significantly in all three dimensions. The intervention group improved significantly compared to the Passive group in all three dimensions, and in the case of OVal-pd concrete and OVal-pd symbolic, it is at the significance level *p* < 0.05. Compared to being passive during the days, this intervention seems to be successful. The clinical significance compared to the controls, measured by Hedges’ g, is approximately 0.60 in all three dimensions, indicating a good effect size.

Theme number two in the interview study is about perceived values in everyday occupations. The participants clearly found increased motivation, self-esteem, joy and empowerment. This theme validates the results of the primary outcome measure OVal-pd. By introducing riding as a new activity in their lives, participants felt that they can cope better in their everyday lives. They claimed that riding was perceived meaningful from several perspectives; that it offered both pleasure and benefit, and they felt they obtain an identity they were proud of: they were not patients, they were riders. Results that show that participants feel that they have been given a different identity, that they have become capable and can master a large horse, are a very important observation and correspond to previous findings [69]. The riding, in the form it is carried out in the project, increases the participants’ occupational values in all three dimensions. The empowerment that this also leads to is a process in which the person’s own resources are mobilized to be able to control their lives, meet their needs and solve their own problems. From a patient perspective, the feeling of empowerment can be strengthened through trust in the staff, the opportunity to learn, and participation in decision-making regarding choice of treatment [70].

In the interview study, theme number one was related to the participants’ awareness that they had improved their health through the intervention. This applied to a wide range of symptoms that had improved or, for symptoms that usually deteriorate due to the progressive nature of the disease, whose deterioration had slowed down: balance, energy, quality of sleep. It also applied to awareness that the intervention both took and gave energy. The quantitative measurements that showed the same thing (EQ-VAS and SMBQ) were hereby validated by the participants’ statements. All dimensions in the occupations give meaning to the participants’ everyday life [71,72], and having a balance between the three OVal-pd dimensions has a positive effect on health [44,73]. The results clearly show that as all three dimensions of perceived activity values increased, so too did the self-assessed global health, measured by EQ-VAS. We find that the participants in the intervention group ranked their health lowest of all three cohorts at baseline. After 12 months, self-rated health was ranked highest by the intervention group. It was an impressive journey they made, which is statistically significant even at the *p* < 0.05 level, both in terms of differences between baseline and follow-up and compared to the control groups. The effect size in relation to the control groups, measured by Hedges’ g, is particularly great after six months, when it was approximately 0.70. The improvement in health may largely be due to the expansion of the activity repertoire, but it may also be due to the interaction with animals being calming, which may be due to the release of oxytocin [74,75] or the particular type of nonverbal communication that occurs between animals and humans [76].

The Shirom-Melamed Burnout Questionnaire (SMBQ) is an instrument primarily intended to measure changes in stress-related burnout. Many participants felt that they were tired and many felt some stress in not being able to cope with everyday life. We therefore chose to use this instrument as a secondary outcome measure. For both groups, an instrument measuring perceived mental fatigue would have been more appropriate to assess the possible effects of the intervention on perceived tiredness. Riding in nature was perceived as calming and restorative, where the participants’ could focus on being here and now. In a recent study, sensory experience in nature promoted mental restoration and perceived improved mental recovery [77,78], which seems to be the case especially when riding in nature.

### 4.1. Issues Regarding Future Policy

Animal-assisted interventions can improve quality of life and activities of everyday life for individuals living with neurological disorders, but have rarely been noted [79]. The intervention has given all participants the opportunity to be able to work towards their own goals, despite varying fundamental problems. Thirty minutes on horseback at walk corresponds to physical exertion equivalent to a half-hour walk. To support health, it is known that a minimum of 150 min of moderate intensive physical activity is recommended per week [80]. Several participants have had a very large reduction in strength and mobility, but have still been able to ride for several years. There are not many other activities that produce the same impact physically, mentally and socially for those who have extensive disabilities. Recent studies have shown that multimodal rehabilitation including horseback riding could lead to long-term physical improvements and may have a profound impact on the emotional state of stroke survivors with moderate levels of disability [81,82]. Our results indicate that this could be the fact even for people with more severe levels of disability and with different neurological disorders.

Two themes in the interview study concerned the conditions for people with neurological disorders to be able to ride on their own. These relate to accessibility and financial conditions. Much deals with how society can contribute in order to allow people with neurological disorders can take part in this activity.

The third theme addresses the vulnerability and helplessness that participants experience. They do not have many resources, whether economic, social or physical. They are aware of how much the intervention means to them, but they do not have the financial resources to be able to pay to get the same service privately. Through the intervention, they are no longer vulnerable, helpless patients but riders, among other riders. They express a feeling of empowerment supported by the perceived significant increased perception of general health (Figure 8). Being in control and being able to perform tasks can contribute to increased quality of life [83]. They hope that politicians will be able to understand the tremendous value, even in savings for society, of the fact that they can strengthen and/or maintain their health through riding.

The fourth theme addresses the prerequisites and obstacles. Participants refer to what is needed in terms of location, staff, horses, premises and nature where they can engage in horseback riding. Requirements must be set higher than those for healthy individuals in order to obtain the expected results. The participants have access to nature that would otherwise be out of reach because of their limitations. The participants describe that getting out in nature and enjoying themselves is an opportunity for mental recuperation. The horse takes the rider along small forest paths, while they can enjoy the environment and focus on the experience. In the stables and riding hall, however, riding becomes more like a workout that requires concentration and focus on the movements.

The national disability policy is currently increasing governance towards full participation and equality in living conditions for people with disabilities [60]. This intervention with equine-assisted activity covering a maximum of 30 min per week affects perceived health, quality of life and occupational value in everyday life. It might be considered as an alternative for improving inclusion and participation for people in a vulnerable situation. An intervention such as this can help to reduce health inequalities and may also help to strengthen the effects of healthcare rehabilitation efforts so that people can maintain their functions, maintain their work ability and delay deterioration despite, in many cases, progressive disease. Future research regarding interventions such as this should aim to gain a deeper understanding from the participants’ point of view, clarify the impact of the different elements, i.e., the horse, the therapist and the environment, and analyze the cost effectiveness of the intervention.

### 4.2. Study Limitations

This study adds new knowledge about how an equine-assisted activity could improve perceived value of everyday occupations and quality of life for people with lifelong neurological disorders. The strengths of the study is that it is prospective, longitudinal, has two control groups and uses triangulation with both quantitative and qualitative methods. However, the Shirom-Melamed Burnout Questionnaire is designed for participants with burnout syndrome. While the research team has conducted studies in the past where we successfully used SMBQ, we suspect that the instrument is not sufficiently sensitive for this group of participants. If we were to conduct this study again, we would choose an instrument more specifically designed for a group of participants with neurological disorders [84]. Another limitation is that the two control groups were not randomized. Thus, we cannot say for sure that they are completely comparable. However, we did check that the outcome variables, age, gender and diagnosis were very similar and that the difference lay in how often they had activities outside the home. Considering that we did not use randomization, we instead chose a method that can be described as a natural experiment or a quasi-experiment [37,38]. Although this could be a limitation, by choosing this option we gain ethically, as well as by research from having followed the groups in a natural way instead of in a randomized controlled trial. The control groups live their lives as usual, where there is no question of any kind of intervention. The risk of dropouts or changes in the control group decreases due to dissatisfaction with not being able to belong to the intervention group and at the same time, one can assume that the values are more similar to those given in ordinary rehabilitation.

## 5. Conclusions

The aim of the study was to investigate the effects of equine-assisted intervention on the participants’ perceived value of everyday occupations and health as well as to gain deeper understanding of what the interventions can mean for participants in everyday life. The Passive control group showed inferior values compared to the Active control group regarding OVal-pd, SMBQ and EQ-VAS. In particular, self-assessed health (EQ-VAS) was clearly higher in the participants of the Active control group. The values regarding the control groups did not change over time. Prior to the start, the intervention group, as well as the Passive control group, had worse values than the Active control group in all areas. As regards how health and everyday activities are estimated, the participants in the intervention group have greatly and significantly improved, so that they pass the Active control group in many areas. Overall, the results indicate that equine-assisted activities as a complement to habilitation and rehabilitation of people with neurological diagnoses can have a positive impact on well-being and global self-assessed health. The primary outcome measure OVal-pd showed that participants increased in all three areas: activities with concrete, symbolic and self-rewarding values. Everyday life has been improved to consist of activities filled with meaning and context: doing something concrete, where everyday life gets structure and content; doing something that gives self-esteem, a symbolic value; doing something of desire, for fun, a self-rewarding value. All of these elements increased considerably for the intervention group, and the increase is statistically significant. Equine-assisted activities have the potential to be health-promoting and a meaningful everyday activity and could be an alternative that not only brings joy and health to the users but could even reduce costs in healthcare. The aspects of decreased health care consumption should be investigated further.

## Figures and Tables

**Figure 1 ijerph-17-02431-f001:**
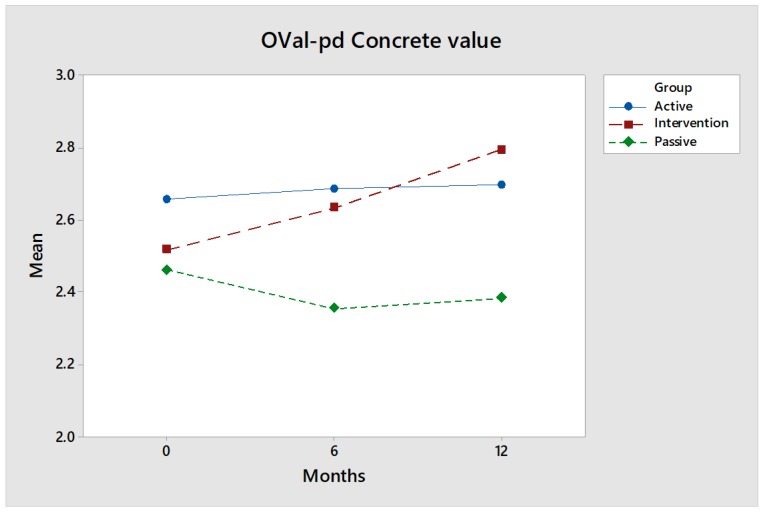
The Oval-pd concrete value for the intervention group (red) increases over time, while the values for the control groups (green and blue) are largely the same.

**Figure 2 ijerph-17-02431-f002:**
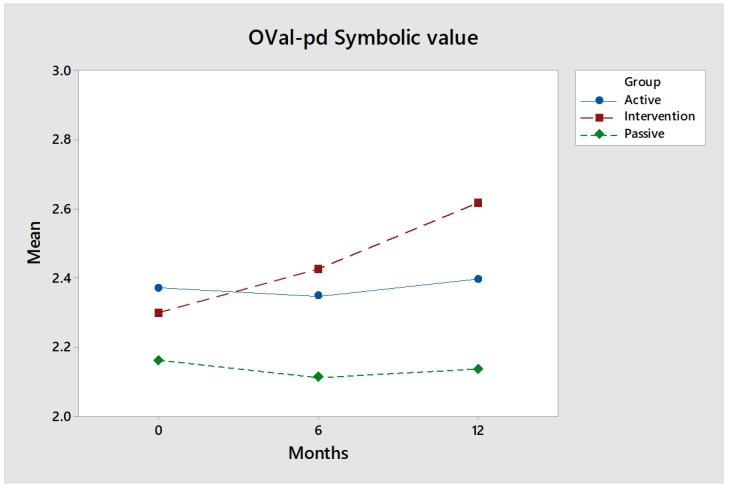
The Oval-pd symbolic value increases over time for the intervention group (red) while the values for the control groups (green and blue) are largely the same over time.

**Figure 3 ijerph-17-02431-f003:**
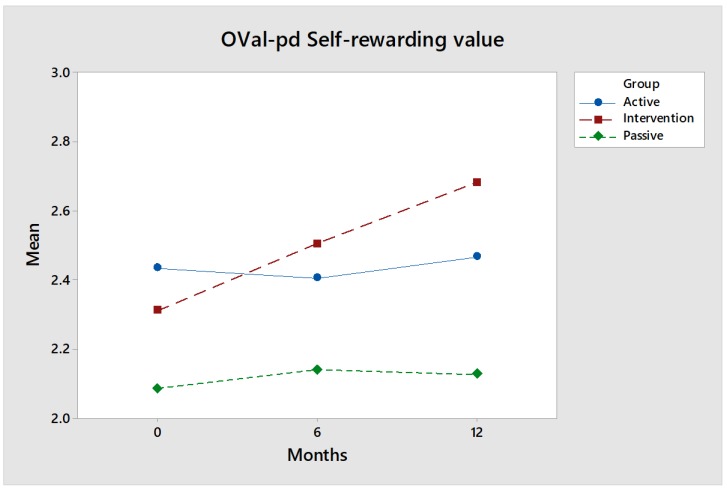
The Oval-pd Self-rewarding value for the intervention group (red) increases over time, while the values for the control groups (green & blue) are largely the same.

**Figure 4 ijerph-17-02431-f004:**
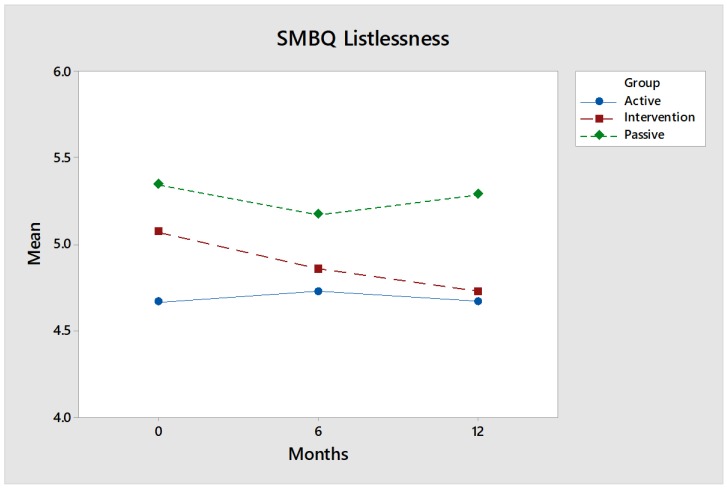
No statistical differences between the intervention group (red) and the control groups (green and blue). However, Shirom-Melamed Burnout Questionnaire (SMBQ) listlessness steadily decreased over time for the intervention group.

**Figure 5 ijerph-17-02431-f005:**
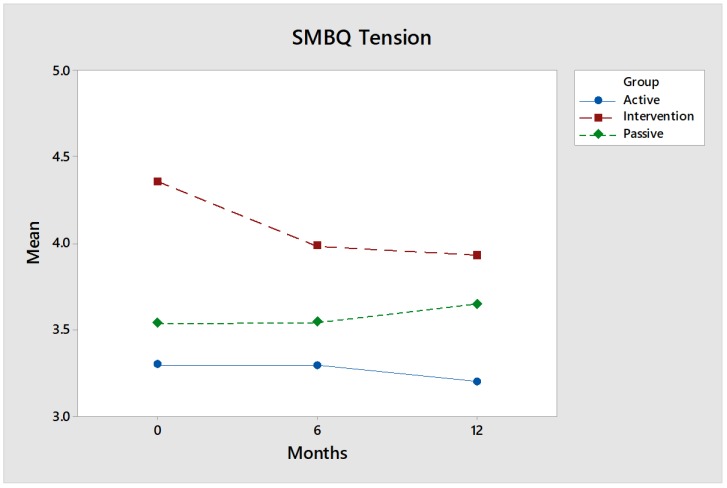
Values of the intervention group (red) improved over time, i.e., participants perceived less SMBQ tension at 12 months, measured by Hedges’ g, which is low, 0.18, for comparisons between the intervention group and the Active control group, but medium, 0.64, for comparisons between the intervention group and the Passive control group.

**Figure 6 ijerph-17-02431-f006:**
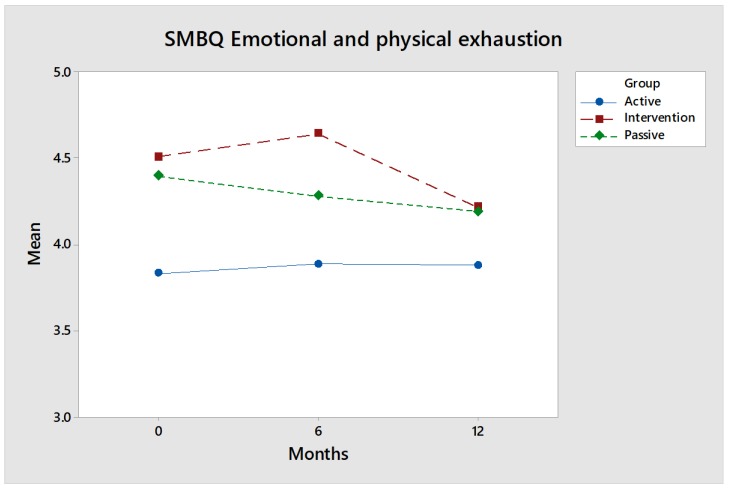
No significant differences over time between the intervention group and the control groups with respect to the dimension SMBQ emotional and physical exhaustion. However, the values of the intervention group are improved between 6 and 12 months.

**Figure 7 ijerph-17-02431-f007:**
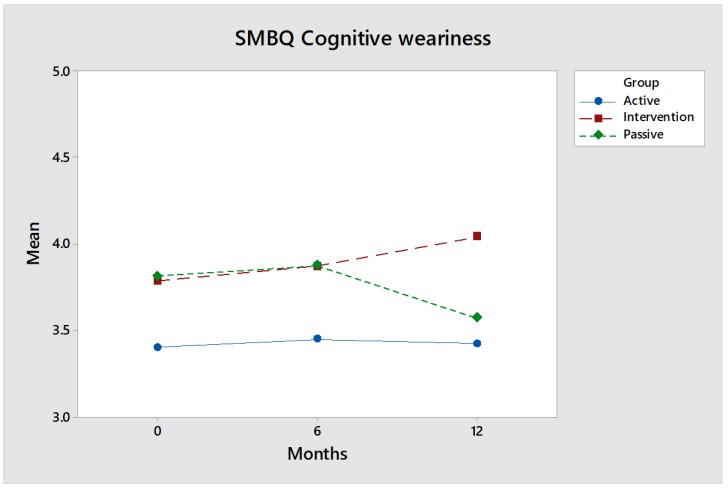
The values regarding SMBQ cognitive weariness for the intervention group (red) increased slightly over time. However, they did not deviate from either baseline or the controls with statistical significance.

**Figure 8 ijerph-17-02431-f008:**
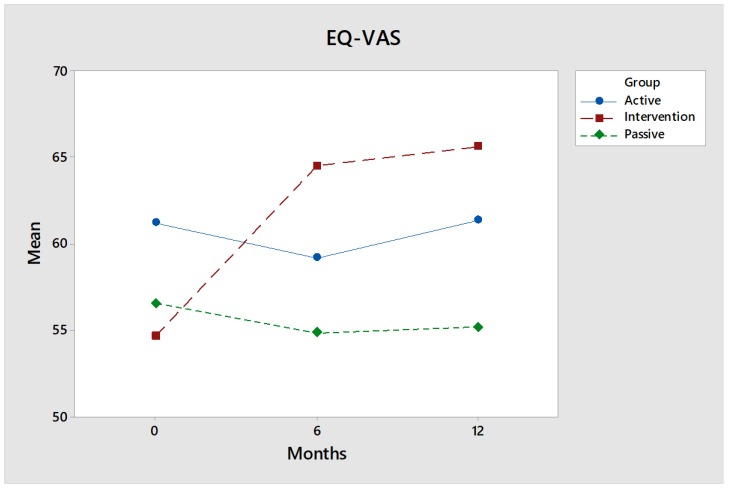
EQ-VAS is a global measure of self-estimated health, where 0 = worst possible health and 100 = best possible health. The average of the intervention group (red) rises significantly over time, while the control groups (green and blue) do not change over time.

**Table 1 ijerph-17-02431-t001:** Description of participants in the study divided by intervention group (*n* = 14) and control group; active (*n* = 147) and passive (*n* = 29).

	Intervention	Active	Passive
Participants	14	147	29
Women (men)	12 (2)	110 (37)	24 (5)
Mean age (distribution)	52 (22–71)	55 (27–84)	51 (19–69)
Active, times/week	1	3.7	0

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
