# Peer review of "Equine-Assisted Intervention to Improve Perceived Value of Everyday Occupations and Quality of Life in People with Lifelong Neurological Disorders: A Prospective Controlled Study"

_ijerph, 2020, doi:10.3390/ijerph17072431_

Round 1
Reviewer 1 Report
The manuscript evaluates a year-long equine-assisted intervention (EAI) for 14 people with neurological disorders. The aim is to investigate possible impact beyond bodily effects, namely, whether the intervention positively affects participants’ activity repertoire, health, and fatigue. Before, after, and half way through the intervention standardised self report measures were used to obtain data from the participants and from two comparison groups with similar neurological issues, and the authors conclude that the intervention group fares better than the comparison groups on most measures. Additionally, themes from interviews with five intervention participants are briefly described and a short summary of the intervention itself is provided.
Thank you for the opportunity to review this manuscript. I agree with the authors that there is a need for interventions for people with neurological disorders that adress their agency also. I also agree that equine assisted activities may have very relevant psycho social effects besides the more well-established somatic effects of hippotherapy, in this and other groups, and that we need more studies of this. Additionally, I would like to applaud the authors for looking into an intervention that runs weekly for a year rather than the 8- og 12- week programs that are more often reported, and I agree that such natural(istic) cases are difficult material for randomized controlled trials. This makes the study/manus worthwhile in itself.
However, I was somewhat confused by a several aspects of the study design and terminology. This is my main concern with the manuscript. I think the authors would make a more convincing case by reconsidering some of the terminology and providing more thorough arguments for some of the choices. I shall mention three elements but I suggest taking a critical look on the methodological terminology and choices in general.
Firstly, the authors describe it as a controlled study with an active and a passive control group. Is it? Are these really control groups, or are they more aptly called comparison groups? Moreover, an active control group usually means a group given the same amount of extra (attention etc.) as the intervention group, in order to control for (be able to subtract) the Hawthorne/placebo-effect. No such control is in this study. We cannot know whether any addition to the participants’ lives would have had the same effect as the EAI, since the EAI group was the only one for whom anything was added. I suggest discussing this in the limitation section, and avoiding the term ”active control group” for what seems to be a non-equivalent comparison group (i.e., those who were habitually more active).
Secondly, I wondered why a measure of burnout was chosen for the dimension of fatigue. Burnout prototypically refers to work-related stress (https://www.who.int/mental_health/evidence/burn-out/en/ ) and I believe there are measures more directly related to the fatigue that is often a side-effect of neurological issues. Since the study is over, it is no use searching for such measures now but the authors’ might supply an argument for choosing a measure from a different domain.
Thirdly, in results I wondered about the choice of significance level p < ,1. That is unusual. Moreover, the figures and text goes on to concern effect sizes so maybe confidence intervals and error bars would be relevant and informative than a lenient p-level.
Minor issues and suggestions:
- I suggest thorough English checking for the final version of the manuscript, and adding English translation of titles when references are in Scandinavian. Some sentences are hard to follow (e.g., line 41-42)
- The article is launched by what is effectively a history section (lines 34-61). This is relevant, I think, but wonder whether a source whose aim is historical analysis of the field wouldn’t be more convincing than claims from a number of secondary sources whose own own aim will have been different. It could also be relevant to refer to the seminal article that notoriously discarded EAT (Anestis et al, 2014) but in effect also provoked an increasing number of studies in the domain that made the field flourish
- Given the focus on increased agency and other positive psycho social sideeffects of EAT for people with disabilities in the present manuscript, the study by Wanneberg (2014) may be a relevant addition to the discussion.
Author Response
Reply to comments from reviewer 1
Reviewer: The manuscript evaluates a year-long equine-assisted intervention (EAI) for 14 people with neurological disorders. The aim is to investigate possible impact beyond bodily effects, namely, whether the intervention positively affects participants’ activity repertoire, health, and fatigue. Before, after, and half way through the intervention standardised self report measures were used to obtain data from the participants and from two comparison groups with similar neurological issues, and the authors conclude that the intervention group fares better than the comparison groups on most measures. Additionally, themes from interviews with five intervention participants are briefly described and a short summary of the intervention itself is provided.
Reply: Clarification: It was not before, after and halfway through the intervention that we collected the self-report measures. On page 5, line 197 we state: “At the start of the intervention as well as after 6 and 12 months of inclusion, the participants in the intervention group had to complete a number of self-assessment forms regarding health and function.”
Reviewer: Thank you for the opportunity to review this manuscript. I agree with the authors that there is a need for interventions for people with neurological disorders that adress their agency also. I also agree that equine assisted activities may have very relevant psycho social effects besides the more well-established somatic effects of hippotherapy, in this and other groups, and that we need more studies of this. Additionally, I would like to applaud the authors for looking into an intervention that runs weekly for a year rather than the 8- og 12- week programs that are more often reported, and I agree that such natural(istic) cases are difficult material for randomized controlled trials. This makes the study/manus worthwhile in itself.
Reply: Thank you! We would also like to thank you for a thorough review of our manuscript.
Reviewer: However, I was somewhat confused by a several aspects of the study design and terminology. This is my main concern with the manuscript. I think the authors would make a more convincing case by reconsidering some of the terminology and providing more thorough arguments for some of the choices. I shall mention three elements but I suggest taking a critical look on the methodological terminology and choices in general. Firstly, the authors describe it as a controlled study with an active and a passive control group. Is it? Are these really control groups, or are they more aptly called comparison groups?
Reply: Thank you for bringing this to our attention; this needs further clarification. This is a controlled study according to, among others, Cochrane. And according to Cochrane, what is important regarding controlled studies is that measurements are taken before, during and after the intervention, in both the intervention group and in control groups. And that is exactly what we did in this study: the groups are constantly followed by the same measurement methods, prospectively. Therefore, this is a prospective controlled study, and more precisely this type of study can be defined as a "Controlled before and after study". This type of study has many advantages regarding internal and external validation. We have added the following on page 2, line 85: “The study was conducted as a prospective controlled before and after study.” References: Cochrane Non-randomized controlled study designs. 2020. https://childhoodcancer.cochrane.org/non-randomised-controlled-study-nrs-designs . Cochrane EPOC resources for review authors. 2020. https://epoc.cochrane.org/resources/epoc-resources-review-authors Grimshaw, J.; Campbell, M.; Ecclesa, M.; Steena, N. 2000. Experimental and quasi-experimental designs for evaluating guideline implementation strategies. Family Practice, 17(S1): 11-16.
In addition, we have added the following on page 2, line 89: “In a case like this, when an RCT was not feasible, we found advantages of including two nonrandomized control groups instead of relying solely on a pre- and post-intervention comparison. Our control groups helped us to account for threats to internal validity (e.g. regression to the mean). The use of a controlled before and after study also reduced threats to external validity, which often limit the value of RCT-studies. Normally RCTs are done at certain highly selected and special settings and are seldom done at ordinary community sites. By using this type of design we could involve common riding schools, and hence making the results more generalizable. The lack of randomization often facilitates recruitment of a larger proportion of eligible participants, thus further increasing generalizability.” Reference: Axelrod, D.A.; Hayward, R. 2007. Nonrandomized interventional study designs, pp 63-76 in: Penson, D.F.; Wei, J.T. (Eds.) Clinical research methods for surgeons. Humane Press: Totowa, N.J.
Reviewer: Moreover, an active control group usually means a group given the same amount of extra (attention etc.) as the intervention group, in order to control for (be able to subtract) the Hawthorne/placebo-effect. No such control is in this study. We cannot know whether any addition to the participants’ lives would have had the same effect as the EAI, since the EAI group was the only one for whom anything was added. I suggest discussing this in the limitation section, and avoiding the term ”active control group” for what seems to be a non-equivalent comparison group (i.e., those who were habitually more active).
Reply: Thank you, this also needs clarification. We have followed Carter & Lubinsky (2016). As we understand, their definition of active and passive control group is the most common. They define "active control group" as participants that receive standard treatment, treatment as usual, or participate in an activity that according to common practice gives effect. The "passive control group" gets placebo or no treatment at all. We have included the following on page 5, line 191: “We define control groups according to Carter & Lubinsky (2016): "active control group" means that participants receive standard treatment, treatment as usual or participate in an activity that according to common practice gives effect. "Passive control group" gets placebo or no treatment at all.” Reference: Carter, R.E. and Lubinsky, J. 2016. Rehabilitation Research: Principles and applications. Elsevier: S:t Louis, Missouri
Reviewer: Secondly, I wondered why a measure of burnout was chosen for the dimension of fatigue? Burnout prototypically refers to work-related stress (https://www.who.int/mental_health/evidence/burn-out/en/ ) and I believe there are measures more directly related to the fatigue that is often a side-effect of neurological issues. Since the study is over, it is no use searching for such measures now but the authors’ might supply an argument for choosing a measure from a different domain.
Reply: It is quite true. We add the following to 4.2 Study limitations, page 19, line 650: "The Shirom-Melamed Burnout Questionnaire is designed for participants with burnout syndrome. The research group has conducted studies in the past where we successfully used SMBQ. However, we suspect that the sensitivity of the instrument is not sufficient for this group of participants. If we had executed this study again, we would have chosen an instrument more specifically designed for a group of participants with neurological disorders.” Reference: Johansson B, Starmark A, Berglund P, Rödholm M, Rönnbäck L. A self-assessment questionnaire for mental fatigue and related symptoms after neurological disorders and injuries. Brain Inj. 2010, 24(1):2-12. doi: 10.3109/02699050903452961.
Reviewer: Thirdly, in results I wondered about the choice of significance level p < ,1. That is unusual. Moreover, the figures and text goes on to concern effect sizes so maybe confidence intervals and error bars would be relevant and informative than a lenient p-level.
Reply: Traditionally it is p<0.05, but it could as well be p<0.20 or p<0.00001. There is no statistical requirement that the significance level should be p <.05. Rather, the level should always be considered, where knowledge of the study participants and the size of the sample are important factors. Having a large sample of 1000 participants, a p-value of <0.05 may arise from sheer coincidence, which should lead to the requirements being set higher, for example at the level p <0.005. Here we have a small sample that we have relatively good knowledge of, with inclusion and exclusion criteria as well as follow-up qualitative studies. On page 7, under Statistical analysis, we write the following: “a significance level of 0.10 was used. In this case, we have good knowledge of all persons included in the study. We also have qualitative analyses that can back up the quantitative methods. The amount of information for each participant is high, which means that we can choose the significance level p <0.10. Where relevant, the significance level p<0.05 is also reported.” We refer to Jae et al (2015) and Perez et al (2014). We also choose to add the references Thiese et al (2016) and Kim and Choi (2019), to support this approach. Reference: Kim, J.H.; Choi, I. 2019. Choosing the Level of Significance: A Decision-theoretic Approach. Abacus: a Journal of Accounting, Finance and Business Studies. doi: 10.1111/abac.12172. Thiese, M.S.; Ronna, B.; Ott, U. 2016. P value interpretations and considerations. J Thorac Dis. 2016 Sep; 8(9): E928–E931. doi: 10.21037/jtd.2016.08.16
In addition, the p-value does not say anything about the effect size. An almost insignificant effect can give a good p-value in really large studies. Looking at weight loss studies, a difference of four kilos can give better p-value in a sample of 500 participants than a sample of 250 participants. In studies with few participants, the effect size is often the most important to convey. Therefore, we choose to report both p-values and effect values. On page 7, under Statistical analysis, we add the following: “P values provide limited information on the clinical efficacy (ref). Therefore, together with the p-values we choose to report the effect size.” References: Dahiru, T. 2008. P-value, a true test of statistical significance? A cautionary note. Ann Ib Postgrad Med. 2008, 6(1): 21–26. Sullivan, G.M.; Feinn, R. 2012. Using Effect Size—or Why the P Value Is Not Enough. J Grad Med Educ. 2012 Sep; 4(3): 279–282. doi: 10.4300/JGME-D-12-00156.1
Minor issues and suggestions:
Reviewer: I suggest thorough English checking for the final version of the manuscript, and adding English translation of titles when references are in Scandinavian. Some sentences are hard to follow (e.g., line 41-42)
Reply: Yes, we are aware of this and will send the manuscript to a professional translator.
Reviewer: The article is launched by what is effectively a history section (lines 34-61). This is relevant, I think, but wonder whether a source whose aim is historical analysis of the field wouldn’t be more convincing than claims from a number of secondary sources whose own own aim will have been different. It could also be relevant to refer to the seminal article that notoriously discarded EAT (Anestis et al, 2014) but in effect also provoked an increasing number of studies in the domain that made the field flourish
Reply: A chapter that gives a good description of the history is the following, which we have included in the background description: Fry N.E. (2013) Equine-Assisted Therapy: An Overview, pp 255-284 in: Grassberger M., Sherman R., Gileva O., Kim C., Mumcuoglu K. (eds): Biotherapy - History, Principles and Practice. Springer, Dordrecht. And thanks for the suggestion of Anestis et al 2014! We have added this reference in the background.
Reviewer: Given the focus on increased agency and other positive psycho social sideeffects of EAT for people with disabilities in the present manuscript, the study by Wanneberg (2014) may be a relevant addition to the discussion.
Reply: Thank you for the suggestion. This is a perfect reference in the discussion, and we have of course added it with the following on page 17, line 566: “Results that show that participants feel that they have been given a different identity, that they have become capable and can master a large horse, is a very important observation and corresponds to previous findings”
Reviewer 2 Report
This experimental study shows results about the effective benefits of Equine Assisted Interventions from a novel perspective, considering both qualitative and quantitative data and, above all, including two control groups. The target are neurological injured people and the study focuses on how their lives can be improved with therapeutic activities involving horses. The group involved in interventions was composed by 14 people; the group of passive people (not performing any activity at all) was composed by 29 people and finally the active group (active one or more time a week) was composed by 147 people. All of them are affected by various neurological disorders. Quantitative analyses have been performed through three different surveys/questionnaires: the OVAL-pd for perceived value of everyday occupation; the SMBQ measuring exhaustion and long-term stress; EQ-VAS for self-assessed health. For qualitative analyses five semi-structured interviews were conducted in the intervention group, with five people from the intervention group. The most interesting results in my opinion regard the rewarding value of the experience (evaluated by OVAL-pd) in which it is clear that people perceived they are doing something in which their skills match the challenge of the activity. Another important accomplishment is that the intervention group was the lowest in evaluating their self-esteemed health at the very beginning and then their rate strongly raised in the follow-ups controls. On the other hand, it is also interesting noting that cognitive weariness was perceived in the intervention group since the activity is really demanding, in both physical and mental terms; this is something which is rarely taken into account. Having say that, I suggest moving Conclusion paragraph (lines 649-667) upwards in the Discussion; in fact, for readers it is important to have the results explained immediately after the Results session.
Major comments: A compelling question is defining the Animal Assisted Interventions and adopting a validated classification of Equine Assisted Interventions. According to the aim of the intervention, they are usually classified in animal assisted therapy (AAT), animal assisted education (AAE) and animal assisted activity (AAA), and they are structured and managed by a multidisciplinary team. This is the classification by the IAHAIO (The International Association of Human-Animal Interaction Organizations, whitepaper updated in 2018). In this huge framework, equine assisted interventions (EAIs) is an umbrella term that includes a wide diversity of methodologies and approaches (Equine assisted therapy—EAT; equine assisted education—EAE; equine assisted activity—EAA). The authors should add the classification they prefer in the manuscript and then better define the interventions they have chosen for the experimental group (EAA, if I understood correctly).
Minor comments: I recommend submitting your manuscript to a native English speaker. However, here are few suggestions.
- Line 35: “not least, relations with horses” this sentence should be included in the previous one and not in the subsequent one
- Line 41-42: I suggest rearranging this sentence, for example like this: “Florence Nightingale was the first therapist which deliberately used animals in a therapeutic treatment during the mid and late 1800s
- Line 58 (and also in many other parts of the manuscript, lines 87, 182, 185): I suggest to not use “it’s about” so often. Maybe a more formal way should be better in a scientific paper. For example, in this case “Therapies with horses were found to improve balance…”
- Line 62: please change in “terms of improvements in both physical and mental health”
- Line 63: remove “that” and add something like “for example” or nothing
- Line 66: change in “control conditions”, change “small” in “few” if you mean not much or “limited” if you mean small in number of subjects involved
- Line 67: change “who do not have” in “who are not affected by/not suffering from”
- Line 91: change “have activities” in “do activities” or “engage in activities”
- Line 101: change “all included participants” in “all participants involved”
- Line 114: what the authors mean with “this goal is met today”? please reformulate
- Line 119: change in “involving physical activities”
- Line 499-500: please reformulate this sentence. It could be like “while at the same time it could be perceived also as a privilege if compared with many other who would have like to and need to be involved”
- Line 512: maybe better than “knowledgeable” could be “competent”
- Line 516: change in “it was nice and soothing spend time with the horse”
- Line 524-525: please reformulate the sentence in a clearer way
- Line 558: I suggest citing Kendall et al., 2013 (Hypotheses about the Psychological Benefits of Horses) or anywhere else the authors think is better.
- Line 567: remove “too”
- Line 576: I suggest citing Scopa et al., 2019 here (Emotional Transfer in Human–Horse Interaction: New Perspectives on Equine Assisted Interventions) and to move citation 57 in line 584 with citations 58 and 59; or anywhere else the authors think is better
- Line 647: change “belong to the intervention group” in “participate in the experimental group” or something similar
- Please explain what IPA is (line 444)
Author Response
Reply to comments from reviewer 2
Reviewer: This experimental study shows results about the effective benefits of Equine Assisted Interventions from a novel perspective, considering both qualitative and quantitative data and, above all, including two control groups. The target are neurological injured people and the study focuses on how their lives can be improved with therapeutic activities involving horses. The group involved in interventions was composed by 14 people; the group of passive people (not performing any activity at all) was composed by 29 people and finally the active group (active one or more time a week) was composed by 147 people. All of them are affected by various neurological disorders. Quantitative analyses have been performed through three different surveys/questionnaires: the OVAL-pd for perceived value of everyday occupation; the SMBQ measuring exhaustion and long-term stress; EQ-VAS for self-assessed health. For qualitative analyses five semi-structured interviews were conducted in the intervention group, with five people from the intervention group. The most interesting results in my opinion regard the rewarding value of the experience (evaluated by OVAL-pd) in which it is clear that people perceived they are doing something in which their skills match the challenge of the activity. Another important accomplishment is that the intervention group was the lowest in evaluating their self-esteemed health at the very beginning and then their rate strongly raised in the follow-ups controls. On the other hand, it is also interesting noting that cognitive weariness was perceived in the intervention group since the activity is really demanding, in both physical and mental terms; this is something which is rarely taken into account.
Reply: Thank you! And thank you for doing such a thorough review of our manuscript!
Reviewer: Having say that, I suggest moving Conclusion paragraph (lines 649-667) upwards in the Discussion; in fact, for readers it is important to have the results explained immediately after the Results session.
Reply: We are hesitant about this and have received judgements of other reviewers who, on the contrary, point out that this is a good location of conclusion.
Reviewer: Major comments: A compelling question is defining the Animal Assisted Interventions and adopting a validated classification of Equine Assisted Interventions. According to the aim of the intervention, they are usually classified in animal assisted therapy (AAT), animal assisted education (AAE) and animal assisted activity (AAA), and they are structured and managed by a multidisciplinary team. This is the classification by the IAHAIO (The International Association of Human-Animal Interaction Organizations, whitepaper updated in 2018). In this huge framework, equine assisted interventions (EAIs) is an umbrella term that includes a wide diversity of methodologies and approaches (Equine assisted therapy—EAT; equine assisted education—EAE; equine assisted activity—EAA). The authors should add the classification they prefer in the manuscript and then better define the interventions they have chosen for the experimental group (EAA, if I understood correctly).
Reply: We will clarify this by adding the following on page 3, 2.2 Intervention: “The intervention we carry out can be defined as "Animal Assisted Activity" according to the International Association of Human-Animal Interaction Organizations (IAHAO) and thus it can be defined as Equine Assisted Activity (EAA) according to affiliated organizations, such as the Professional Association of Therapeutic Horsemanship International (PATH International)”. And we refer to IAHAO white paper.
Reviewer: Minor comments: I recommend submitting your manuscript to a native English speaker.
Reply: We will send the manuscript to a professional translator. Thank you for the suggestions. We will attach your suggestions below to the translator.
However, here are few suggestions.
- Line 35: “not least, relations with horses” this sentence should be included in the previous one and not in the subsequent one
- Line 41-42: I suggest rearranging this sentence, for example like this: “Florence Nightingale was the first therapist which deliberately used animals in a therapeutic treatment during the mid and late 1800s
- Line 58 (and also in many other parts of the manuscript, lines 87, 182, 185): I suggest to not use “it’s about” so often. Maybe a more formal way should be better in a scientific paper. For example, in this case “Therapies with horses were found to improve balance…”
- Line 62: please change in “terms of improvements in both physical and mental health”
- Line 63: remove “that” and add something like “for example” or nothing
- Line 66: change in “control conditions”, change “small” in “few” if you mean not much or “limited” if you mean small in number of subjects involved
- Line 67: change “who do not have” in “who are not affected by/not suffering from”
- Line 91: change “have activities” in “do activities” or “engage in activities”
- Line 101: change “all included participants” in “all participants involved”
- Line 114: what the authors mean with “this goal is met today”? please reformulate
- Line 119: change in “involving physical activities”
- Line 499-500: please reformulate this sentence. It could be like “while at the same time it could be perceived also as a privilege if compared with many other who would have like to and need to be involved”
- Line 512: maybe better than “knowledgeable” could be “competent”
- Line 516: change in “it was nice and soothing spend time with the horse”
- Line 524-525: please reformulate the sentence in a clearer way
- Line 558: I suggest citing Kendall et al., 2013 (Hypotheses about the Psychological Benefits of Horses) or anywhere else the authors think is better.
- Line 567: remove “too”
- Line 576: I suggest citing Scopa et al., 2019 here (Emotional Transfer in Human–Horse Interaction: New Perspectives on Equine Assisted Interventions) and to move citation 57 in line 584 with citations 58 and 59; or anywhere else the authors think is better
- Line 647: change “belong to the intervention group” in “participate in the experimental group” or something similar
Reviewer: Please explain what IPA is (line 444)
Reply: We will replace the abbreviation (IPA) with Interpretative Phenomenological Analysis, and the reference.
Reviewer 3 Report
Review Feedback;
“Equine-assisted intervention to improve perceived value of everyday occupations and quality of life in people with lifelong neurological disorders. A prospective controlled study”
Overview
Thank you for the opportunity to review a very salient and interesting paper. This is an important subject area and one that desperately requires more robust investigative analysis. I commend the authors of this study for a prospective, controlled study, in addition to clear awareness of the limitations of the study (as discussed later in the manuscript).
The study design, review, analysis and discussion are generally appropriate and detailed.
I commend the use of qualitative AND quantitative consideration of this study – this makes this study unique to my knowledge as one considering both aspects. For this field of study, this is important as there are certainly aspects of animal-assisted interventions that can never be fully quantified by typical analytical methods, and as such, qualitative consideration is critical to further dissect the possible benefits of AAI (or in this case, equine assisted interventions).
Please note – I have not commented on specific suggested edits line by line. I am afraid that personal workload and time for review turnaround necessitated a global review and comment on this manuscript instead. However, I will draw attention to specific matters by line number where appropriate.
Specific Comments
Abstract – clear, concise and a neat precis of the paper. I consider this to be appropriate and details the key information, as well as highlighting the suitability of this paper for this journal in my opinion.
Introduction – appropriate coverage of the subject, although the history of equine assisted therapy (EAT) could have reference to the typical range of studies currently published (typically small group, retrospective) and the serious limitations of these for supporting EAT. In addition, I would suggest noting the wider issue of robust data relating to AAI is also lacking and has particular limitations, not least of which is the ethical and possible welfare issues for the animals involved. While the writing style is generally good, there are lapses in clarity and language use. I would suggest serious and tight editing to rectify these as they can inhibit readability and affect clarity of content. Line 41 refers to Florence Nightingale as first AAI therapist – I would review this statement – she was possibly more the first to officially record the value of AAI in her book, Notes on Nursing and also undertook AAI with her owl. There were certainly “therapists” described pre-Nightingale. Line 61 – clarify and define acronym on first use. I would like to see more introduction of some of the questionnaire tools utilised in this study described in the introduction, rather than more detail in the materials and methods – perhaps consider editing some M and M content to the introduction to ensure the reader is appropriately primed and aware. However, overall, the introduction is appropriate for the following study and considers key review areas.
Aim – clear and appropriate
Materials and Methods – I initially confess to being quite anxious that this study has no specific ethical approval and would strongly suggest that the authors obtain and clarify rather more detail about the lack of formal ethical governance. In my experience of studies like this (especially as a prospective study), then ethical review is still undertaken, as participants could be deemed as vulnerable. I appreciate that written consent was obtained, but that differs from external review and governance of possible ethical infringements. I would urge editors to consider this also (line 95-101) for acceptance of this paper.
Overall, Materials and Methods are descriptive and appropriate. Language use does need improved clarity and tenses (mixing past and present) need editing and consistency. Some content needs introducing in the Introduction to ensure flow of content (as detailed earlier) – this especially applies to some of the descriptors of the survey tools utilised.
Lines 151-161 – how consistent were exercises undertaken? Or was there huge variation? What was the level of indoor versus outdoor riding? I ask as evidence suggests that exposure to the outdoors via riding can be significant beyond the equine aspect (see E.O. Wilson and Biophilia for example) – it would be worthy to note this (is possible).
Interested in the gender differences in the study population – this could be significant, although opportunistic recruitment limits opportunities to balance gender profiles. It should be noted (esp. later in discussion) the possible impacts of gender on outcomes.
Lines 219-229 – some info could be moved to into OR presented as a table for clarity and easy access?
Lines 231-251 – as above
Lines 252-259 – as above
(However, the use of these tools is well employed and described)
Line 260-271 – statistical analysis; appears appropriate although significance level and effect size could be further detailed and the relative outcome benefits of each discussed.
Results – clearly structured and presented. Some language use, sentence and paragraph structure does need editing for clarity.
Figures – no error bars. Possibly entirely appropriate given the nature of the data but could be useful and further illustrate data viability? Please reconsider colours of lines – especially red and green for accessibility – especially important for a paper covering this subject area! Query if printing will facilitate use of colours also?
Line 321 – P slightly above 0.05 – not really advised – within realms of “trends towards”
Supporting descriptive text for figures is clear and assists clarity, however.
Line 443 onwards - Qualitative analysis is well detailed and described but does lack specifics (level to which each theme was met within the intervention group? Any specific demographic differences worthy of note? I appreciate anonymisation but this could be critical to further understanding the value of the intervention for specific demographic groups?) Also, review tenses used and language clarity.
Discussion – nice first sentence BUT is a single sentence paragraph! Consider adding a sentence or two of review of study aim to aid clarity of subsequent discussion for reader.
Good consideration of study group sizes and limitations.
Line 547 – Clarify and edit reference to “A theme (no two)” – similar comment made about reference to “one theme” in line 559
Line 556 – key point made about empowerment – this could be further discussed and has been noted in many EAT studies. Suggest this is a key area worthy of reviewing and discussion further!
Line 601-604 – I think it is important to note that studies such as this one are ESSENTIAL to promote the acceptance and value of interventions such as EAT. Only with robust evidence can such therapies be increasingly accepted by the medical profession and thus also attract funding. However, the importance of goal setting and outcome-based therapies is also critical – query if specific outcomes were discussed with participants? This could be a useful adjunct in future studies, to review potential for measuring improvements in motor function etc.
Limitations – appropriate and reflective.
Conclusions – suggest adding an intro sentence to restate aim of study for clarity. Edit and ensure clarity of wording and sentence structure e.g. line 661-662 – fragmented.
Thank you again for the opportunity to review this paper. I trust you will take these comments in the supportive and reflective manner in which they are intended, as this is a largely very robust and extremely interesting study, that adds to this field significantly.
Author Response
Reply to comments from reviewer 3
Reviewer: Thank you for the opportunity to review a very salient and interesting paper. This is an important subject area and one that desperately requiares more robust investigative analysis. I commend the authors of this study for a prospective, controlled study, in addition to clear awareness of the limitations of the study (as discussed later in the manuscript).
The study design, review, analysis and discussion are generally appropriate and detailed.
I commend the use of qualitative AND quantitative consideration of this study – this makes this study unique to my knowledge as one considering both aspects. For this field of study, this is important as there are certainly aspects of animal-assisted interventions that can never be fully quantified by typical analytical methods, and as such, qualitative consideration is critical to further dissect the possible benefits of AAI (or in this case, equine assisted interventions).
Please note – I have not commented on specific suggested edits line by line. I am afraid that personal workload and time for review turnaround necessitated a global review and comment on this manuscript instead. However, I will draw attention to specific matters by line number where appropriate.
Reply: Thanks, and thank you for your time and energy to do a thorough review!
Reviewer: Specific Comments. Abstract – clear, concise and a neat precis of the paper. I consider this to be appropriate and details the key information, as well as highlighting the suitability of this paper for this journal in my opinion.
Introduction – appropriate coverage of the subject, although the history of equine assisted therapy (EAT) could have reference to the typical range of studies currently published (typically small group, retrospective) and the serious limitations of these for supporting EAT. In addition, I would suggest noting the wider issue of robust data relating to AAI is also lacking and has particular limitations, not least of which is the ethical and possible welfare issues for the animals involved.
Reply: The arguments can preferably be strengthened regarding animal ethics. We add the following. “In addition, animal ethics are all too often overlooked.” References: De Santis, M.; Contalbrigo, L.; Borgi, M.; Cirulli, F.; Luzi, F.; Redaelli, V.; Stefani, A.; Toson, M.; Odore, R.; Vercelli, C.; Valle, E.; Farina, L. Equine Assisted Interventions (EAIs): Methodological Considerations for Stress Assessment in Horses. Vet. Sci. 2017, 4, 44; doi:10.3390/vetsci4030044. Ng, Z.; Albright, J.; Fine, A.H.; Peralta, J. Our Ethical and Moral Responsibility: Ensuring the Welfare of Therapy Animals, pp 357-376 in: Fine, A.H. (Ed). Handbook on Animal-Assisted Therapy. Foundations and Guidelines for Animal-Assisted Interventions. 4th ed. 2015, Academic Press: Cambridge, Massachusetts
Reviewer: While the writing style is generally good, there are lapses in clarity and language use. I would suggest serious and tight editing to rectify these as they can inhibit readability and affect clarity of content.
Reply: Yes, we are well aware of this and will send the manuscript to a professional translator.
Reviewer: Line 41 refers to Florence Nightingale as first AAI therapist – I would review this statement – she was possibly more the first to officially record the value of AAI in her book, Notes on Nursing and also undertook AAI with her owl. There were certainly “therapists” described pre-Nightingale.
Reply: Thank you, we have reviewed the statement according to your suggestion.
Reviewer: Line 61 – clarify and define acronym on first use.
Reply: We have added what the abbreviation means according to your suggestion: heart rate variability.
Reviewer: I would like to see more introduction of some of the questionnaire tools utilised in this study described in the introduction, rather than more detail in the materials and methods – perhaps consider editing some M and M content to the introduction to ensure the reader is appropriately primed and aware. However, overall, the introduction is appropriate for the following study and considers key review areas.
Reply: Primary outcome is about the everyday activity repertoire. We can agree that this issue needs to be primed in the introduction. We choose to do this with the following: “However, health and well-being are not just about symptoms associated with the neurological diseases per se. In general, it is about not being isolated in time and space, about being able to have social contacts and about getting structure in everyday life. It's about being able to have an everyday life that contains activities that bring meaning and context in life. Whether equine assisted activities can influence this, can be partly investigated through interviews and partly by measuring participants' everyday activity repertoire.” References: Gallagher, M.B.; Orla T. Muldoon, O.T.; Pettigrew, J. An integrative review of social and occupational factors influencing health and wellbeing. Front. Psychol., 2015, https://doi.org/10.3389/fpsyg.2015.01281 Schwanen, T.; Wang, D. 2014 Well-Being, Context, and Everyday Activities in Space and Time, Annals of the Association of American Geographers, 104:4, 833-851, DOI: 10.1080/00045608.2014.912549
Reviewer: Aim – clear and appropriate. Materials and Methods – I initially confess to being quite anxious that this study has no specific ethical approval and would strongly suggest that the authors obtain and clarify rather more detail about the lack of formal ethical governance. In my experience of studies like this (especially as a prospective study), then ethical review is still undertaken, as participants could be deemed as vulnerable. I appreciate that written consent was obtained, but that differs from external review and governance of possible ethical infringements. I would urge editors to consider this also (line 95-101) for acceptance of this paper.
Reply: We find that we have to reformulate this paragraph, because we have followed all ethical review procedures. We submitted the study for ethical review at Lund University and followed their advice. The Ethics Review Board in Lund considered this was not a clinical study. That is because the intervention included adults who participated in leisure riding as equine-assisted activities, and hence, there was no dependency or power relationship between participants and researchers regarding treatment, therapy or care. However, they advised us to provide all participants with informed consent in accordance with the WMA Declaration of Helsinki. Among other things, the participants were informed that they could cancel their participation at any time if they no longer wanted to participate, without having to give reasons. We have written the following: “We submitted the study for ethical review at Lund University. They advised us to provide all participants with informed consent in accordance with the WMA Declaration of Helsinki, which we did. Among other things, the participants were informed that they could cancel their participation in the study at any time if they no longer wanted to participate, without having to give reasons. Each participant was asked for their written consent to participate in the study, which was given from all included participants.”
Reviewer: Overall, Materials and Methods are descriptive and appropriate. Language use does need improved clarity and tenses (mixing past and present) need editing and consistency. Some content needs introducing in the Introduction to ensure flow of content (as detailed earlier) – this especially applies to some of the descriptors of the survey tools utilised.
Reply: As stated above, we are well aware of that, and will send the manuscript to a professional translator.
Reviewer: Lines 151-161 – how consistent were exercises undertaken? Or was there huge variation? What was the level of indoor versus outdoor riding? I ask as evidence suggests that exposure to the outdoors via riding can be significant beyond the equine aspect (see E.O. Wilson and Biophilia for example) – it would be worthy to note this (is possible).
Reply: We have added the following: “The exercises were reasonably consistent but were adapted to the different participants' daily state and fitness. Most often, the riding took place indoors, but when the weather allowed, it was moved outdoors. What was defined as "good weather" depended on the participants. Some felt better when it was warm, others when it was cooler.”
Reviewer: Interested in the gender differences in the study population – this could be significant, although opportunistic recruitment limits opportunities to balance gender profiles. It should be noted (esp. later in discussion) the possible impacts of gender on outcomes.
Reply: This intervention attracts more women than men, which reflects equestrian sport in general. It is female-dominated as an everyday activity, but it is a more even gender distribution in competition sports.
Reviewer: Lines 219-229 – some info could be moved to into OR presented as a table for clarity and easy access?
Lines 231-251 – as above
Lines 252-259 – as above
(However, the use of these tools is well employed and described)
Reply: We choose to keep it in its present shape.
Reviewer: Line 260-271 – statistical analysis; appears appropriate although significance level and effect size could be further detailed and the relative outcome benefits of each discussed.
Reply: We have added some information about p-values and effect size, and added some references.
Reviewer: Results – clearly structured and presented. Some language use, sentence and paragraph structure does need editing for clarity.
Reply: Yes, and as stated above, we are well aware of that, and will send the manuscript to a professional translator.
Reviewer: Figures – no error bars. Possibly entirely appropriate given the nature of the data but could be useful and further illustrate data viability? Please reconsider colours of lines – especially red and green for accessibility – especially important for a paper covering this subject area! Query if printing will facilitate use of colours also?
Reply: The article will be published online. The fact that the lines in the figures are in color is therefore of no concern
Reviewer: Line 321 – P slightly above 0.05 – not really advised – within realms of “trends towards” Supporting descriptive text for figures is clear and assists clarity, however. Line 443 onwards - Qualitative analysis is well detailed and described but does lack specifics (level to which each theme was met within the intervention group? Any specific demographic differences worthy of note? I appreciate anonymisation but this could be critical to further understanding the value of the intervention for specific demographic groups?) Also, review tenses used and language clarity.
Reply: The themes described were very clear and can thus be said to be well met within the intervention group. There were no specific demographic differences worth noting. We will have a translator review the entire manuscript.
Reviewer: Discussion – nice first sentence BUT is a single sentence paragraph! Consider adding a sentence or two of review of study aim to aid clarity of subsequent discussion for reader. Good consideration of study group sizes and limitations.
Reply: Again, we are well aware of that, and will send the manuscript to a professional translator.
Reviewer: Line 547 – Clarify and edit reference to “A theme (no two)” – similar comment made about reference to “one theme” in line 559
Reply: Thank you; we have changed this to "theme number two" and "theme number one" respectively
Reviewer: Line 556 – key point made about empowerment – this could be further discussed and has been noted in many EAT studies. Suggest this is a key area worthy of reviewing and discussion further!
Reply: We have added more to this paragraph, as well as a reference highlighting this.
Reviewer: Line 601-604 – I think it is important to note that studies such as this one are ESSENTIAL to promote the acceptance and value of interventions such as EAT. Only with robust evidence can such therapies be increasingly accepted by the medical profession and thus also attract funding. However, the importance of goal setting and outcome-based therapies is also critical – query if specific outcomes were discussed with participants? This could be a useful adjunct in future studies, to review potential for measuring improvements in motor function etc.
Reply: We have included the following under the heading 2.2 Intervention: “Throughout the intervention, the leaders follow the participants' individual goals for the activity”
Reviewer: Limitations – appropriate and reflective. Conclusions – suggest adding an intro sentence to restate aim of study for clarity. Edit and ensure clarity of wording and sentence structure e.g. line 661-662 – fragmented.
Reply: We have added an intro sentence to restate the aim of the study.
Reviewer: Thank you again for the opportunity to review this paper. I trust you will take these comments in the supportive and reflective manner in which they are intended, as this is a largely very robust and extremely interesting study, that adds to this field significantly.
Reply: Thank you very much for your valuable comments.
Round 2
Reviewer 3 Report
Thank you for your responses and detailed amends.
I am happy to support acceptance of the paper in this form with the only tiny stipulation being to add the acronym "RCT" beside "Randomised Controlled Trials" in line 93 for full clarity.
Thank you again.